# Propolis as a Key Source of *p*-Coumaric Acid Permeating Honey and Sucrose Syrup Stores of Honey Bees

**DOI:** 10.3390/insects16111159

**Published:** 2025-11-13

**Authors:** Petra Urajová, Václav Krištůfek, Alena Krejčí

**Affiliations:** 1Centre Algatech, Institute of Microbiology, Czech Academy of Sciences, 379 81 Třeboň, Czech Republic; urajova@alga.cz; 2Biology Centre, Institute of Soil Biology, Czech Academy of Sciences, 370 05 České Budějovice, Czech Republic; vaclav.kristufek@upb.cas.cz; 3Biology Centre, Institute of Entomology, Czech Academy of Sciences, 370 05 České Budějovice, Czech Republic; 4Faculty of Science, University of South Bohemia, 370 05 České Budějovice, Czech Republic

**Keywords:** honey bee, phenolic compounds, *p*-coumaric acid, honey bee nutrition, propolis, honey, supplementary feeding

## Abstract

Beekeepers feed their bees with supplementary saccharide feeding (such as sugar syrup, corn syrup or invert) when honey is harvested from the hive or when natural resources are scarce. These supplementary feeds, however, do not have the same nutritional composition as honey. One missing group of nutrients are phenolic acids that help bees cope with stress, enhance their immunity and prolong life span. In our study, we measured one of these compounds, *p*-coumaric acid, in honey, other bee products and sugar syrup (sucrose solution). At the beginning, the sugar syrup did not contain this compound. However, once bees stored it in their combs, the syrup contained similar levels of *p*-coumaric acid as honey. We discovered that this happens because propolis—a resinous substance bees make from plant resins and their own secretions, used to spread inside the hive, including the surfaces of the combs—releases *p*-coumaric acid into the stored syrup and honey. Other beneficial substances from propolis may enrich the bees’ stored food in the same way. These findings highlight the important role of propolis in bee nutrition and show that supplementary feeding does not necessarily deprive honey bees of vital phytochemicals like phenolic acids.

## 1. Introduction

The naturally occurring phenolic compound *p*-coumaric acid demonstrates a broad range of biological activities across taxa. In humans, it exhibits antioxidant, anti-inflammatory, antimicrobial and anticancer properties, contributing to the prevention of cardiovascular, neurodegenerative and oncological diseases [1,2,3]. Honey bee products (such as honey, bee bread or propolis), together with cereal bran and pigmented fruits (e.g., red grapes, cherries, blueberries), represent some of the richest reservoirs of this compound [4].

For honey bees (*Apis mellifera*), *p*-coumaric acid is essential to their health and colony sustainability. It supports detoxification pathways [5], enhances immune responses [6], extends lifespan [7], reduces pathogen load [8] and modulates gene expression linked to caste differentiation [9]. Honey and pollen are regarded as the primary nutritional sources for honey bees rich in *p*-coumaric acid. The origin of *p*-coumaric acid in pollen is well established, as it is a major component of the sporopollenin matrix of pollen walls [10]. However, the source of *p*-coumaric acid found in honey remains unclear, given its negligible concentration in nectar [11,12,13].

While honey and pollen contain significant amounts of *p*-coumaric acid [14,15,16], the highest concentrations within the hive are reported in propolis [17,18]—a resinous material produced by bees from plant material that is used to coat inner hive surfaces, thereby protecting the colony from bacterial and fungal pathogens. Data from direct measurements of *p*-coumaric acid content in worker diets and wax combs are lacking. Reported concentrations of *p*-coumaric acid in honey, pollen and propolis vary among studies, likely reflecting differences in their geographical origin and analytical methodologies [14,15,16,17,18]. This highlights the need for a direct comparison of *p*-coumaric acid levels across various bee products, using a standardized approach.

A widespread beekeeping practice involves the replacement of harvested honey with saccharide-rich substitutes, most commonly sucrose syrup, fructose corn syrup or invert [19]. While studies have found no significant differences in overall colony winter survival between those fed honey and those fed sucrose syrup [20,21,22], bees overwintering on their own honey tend to exhibit enhanced physiological traits, including increased fat body reserves, increased lifespan and gene expression profiles associated with stress resilience [22,23]. Additionally, bees fed with honey show improved cognitive function and distinct gut microbiota profiles compared to those maintained on a sucrose diet [24].

These findings suggest that supplemental saccharide feeds may lack nutritional components present in honey. The physiological disparities observed between bees fed honey and those fed artificial saccharide sources likely reflect the complex composition of honey that includes numerous bioactive compounds [25]. Notably, phenolic acids such as *p*-coumaric acid, caffeic acid, vanillic acid and chlorogenic acid may contribute to the health promoting effects of natural honey [26,27]. Consequently, concerns have been raised that sucrose syrup supplementation may deprive bees of these phytochemicals essential for coping with environmental and physiological stress [5,22].

In this study, we analyzed the *p*-coumaric acid content in honey and other honey bee products as well as in sucrose syrup before and after feeding to bees. Our goal was to compare *p*-coumaric acid levels in various hive-derived matrices using a standardized HPLC-HRMS approach, to investigate the origin of *p*-coumaric acid in honey and to assess whether winter feeding of sucrose syrup compromises *p*-coumaric acid availability in bee diets, potentially affecting colony health and survival.

## 2. Materials and Methods

### 2.1. Sources of Honey Bee Colonies and Honey Bee Products

Samples of honey, sucrose syrup stores, combs, queen cells, royal jelly, worker larval diet and propolis were taken from seven different apiaries in the region of South Bohemia, Czech Republic (the location of apiaries can be found in Appendix A). The colonies of *Apis mellifera carnica* were managed according to standard practice, that included anti-varroa treatment (amitraz fumigation in autumn, oxalic acid drops in winter, formic acid in summer), honey harvest in June, and sucrose syrup feeding in August. The levels of Varroa destructor were regularly monitored by alcohol washes and did not exceed 2% levels throughout the season.

Honey samples were harvested in July 2021 and July 2022 and honey bee colonies were then fed with sucrose syrup in August, according to a typical supplementary winter feeding protocol routinely used by the majority of bee keepers in the country. This comprised three doses of 5 kg granulated sucrose diluted with 3 L of tap water, offered in one-week intervals using bucket feeders placed on top of the frames inside an additional empty super (in total 15 kg sucrose for each colony was fed within two weeks). For the first experiment performed in August 2021 (data in Figure 2A), empty combs originally built from wax foundations were labelled by permanent marker and inserted in the upper super of the hive on the same day as the supplementary feeding with sucrose syrup started, to avoid any honey deposition instead of the sucrose syrup stores. The labelled capped combs full of sucrose syrup stores were left in the colonies until February 2022 when they were removed, samples of the sucrose stores were carefully extracted from the comb cells by a spatula and the levels of *p*-coumaric acid were assessed in the same run as samples of honey extracted from the same colonies. In the second experiment conducted in August 2022 (data in Figure 2B), empty frames were inserted in the same manner as described above. The capped sucrose syrup stores were analyzed two weeks after the supplementary feeding and then stored outside the hive at room temperature in the laboratory until subsequent analysis in November 2022 and February 2023. In both experiments, three samples of sucrose stores originating from different parts of the combs were analyzed.

Royal jelly and worker larval diet samples were collected from queen and worker brood cells, respectively, using sterile spatulas immediately after larval removal. Corbicular pollen was obtained using pollen traps placed at hive entrances, while bee bread was harvested from comb cells following comb freezing to facilitate extraction. For the analysis of *p*-coumaric acid content in empty storage or brood combs, three samples were collected from different parts of each comb.

### 2.2. Analysis of p-Coumaric Acid Content in Honey, Sucrose Syrup Stores and Royal Jelly

Extraction of *p*-coumaric from these matrices was based on [28]. 1 g of matrix was dissolved in 10 mL of water acidified with 35% hydrochloric acid (Avantor, Radnor, PA, USA) to pH = 2. After activating SPE (solid phase extraction) cartridge (Chromservis, Prague, Czech republic) by methanol and washing with water acidified with HCl to pH = 2, the dissolved sample was loaded. The cartridge was then washed with 6 mL of water acidified with HCl to pH = 2. The *p*-coumaric acid was eluted by 6 mL of methanol. The eluent was concentrated under vacuum and subjected to HPLC-HRMS (HPLC-Thermo Scientific, Sunnyvale, CA, USA, HRMS- Bruker, Billerica, MA, USA) analysis. The concentration of *p*-coumaric acid was normalized to wet weight. Error bars in graphs represent standard deviations of biological triplicate measurements.

### 2.3. Analysis of p-Coumaric Acid in Empty Wax Combs, Queen Cells, Corbicular Pollen, Bee Bread and Propolis

Extraction of *p*-coumaric from these matrices was performed according to a modified method of Mohdaly et al. [29]. Methanol extraction from these matrices proved more effective than water, with maximal compound recovery achieved after one to two weeks of extraction (Appendix A). One gram of matrix was dispersed in 1 mL of methanol (Avantor, Radnor, PA, USA). After vortexing and ultrasonicating the suspension, samples were macerated for two weeks with occasional shaking. After centrifugation, the supernatant was subjected to HPLC-HRMS analysis. The concentration of *p*-coumaric acid was normalized to wet weight. Error bars in graphs represent standard deviations of biological triplicate measurements.

Empty storage combs were built in the same season they were analyzed, either with or without the use of wax foundations. They had not been used by the bees for brood rearing, nor for storage of saccharide stores and pollen. Tweezers were used to remove propolis rings. Empty brood combs were combs previously used for brood rearing. Their colour reflected both age and intensity of brood rearing, with darker combs indicating older and/or more intensively used combs. Prior to use, the combs were inspected to ensure the area sampled for *p*-coumaric acid content was free of honey or pollen. Frames were removed from colonies and stored at 8 °C until analysis.

For analysis of *p*-coumaric acid in queen cells, fifteen empty queen cells (within 24 h after queen emergence) were collected from different hives, divided into light, medium dark and dark categories and each queen cell was analyzed separately for its content of *p*-coumaric acid

### 2.4. The HPLC-HRMS Analyses

Samples for high performance liquid chromatography—high resolution mass spectrometry were analyzed on Dionex UltiMate 3000 UHPLC+ (Thermo Scientific, Sunnyvale, CA, USA) equipped with a diode-array detector. Separation of compounds was performed on reversed-phase C18 column (Phenomenex Kinetex, 150 × 4.6 mm, 2.6 µm, Torrance, CA, USA) using H_2_O (A) and Methanol (B) (both containing 0.1% HCOOH) as a mobile phase at a flow rate of 0.6 mL min^−1^. The gradient was as follows: A/B 70/30 (0 min), 70/30 (in 1 min), 0/100 (in 20 min), 0/100 (in 30 min), and 70/30 (in 35 min). The mass spectrometer operated in negative ionization with the following settings: dry temperature 250 °C; drying gas flow 10 L·m^−1^; nebulizer 3 bar; capillary voltage 3500 V; endplate offset 500 V. The spectra were collected in the range 100–1000 m/z with spectra rate 3 Hz. Sodium formate clusters were used as a calibrant at the beginning of each analysis.

### 2.5. Content of p-Coumaric Acid Calculation

The concentration of *p*-coumaric acid in the samples was determined using an external calibration curve constructed from a commercially available *p*-coumaric acid standard (Merck, Darmstadt, Germany). Extracted ion chromatograms (EICs) corresponding to the protonated molecular ion [M + H]^+^ of *p*-coumaric acid (C_9_H_8_O_3_) were obtained for both standard solutions and sample extracts. In addition, due to in-source decarboxylation observed during ionization, a second EIC corresponding to the decarboxylated form (C_8_H_8_O) was evaluated. For quantification, sum of peak areas from both the sample EICs were compared to those of the standard, and only peaks with retention times matching those of the *p*-coumaric acid standard were considered.

### 2.6. Propolis Solubility in Sucrose Syrup and Honey

Propolis derived from hives was homogenized into powder using a blender. 150 mg of propolis was weighed into separate 2 mL Eppendorf tubes, mixed with 2.5 g honey or sucrose syrup (1:1 ratio sucrose to water) and incubated in a dark incubator at 35 °C for 1 week to 3 months with occasional inverting of tubes. Honey with a low basal level of *p*-coumaric acid was selected for the experiment. On the day of harvesting, samples with sucrose syrup were centrifuged at 10,000× *g* for 5 min and filtered through a 10 µm PSF membrane filter (Pall Corporation, New York, NY, USA) to eliminate unsolubilized propolis particles. Samples containing honey were diluted with the same volume of water just before filtration and processed immediately the same way as samples with sucrose syrup. Content of *p*-coumaric acid was measured as described above and normalized to the weight of honey or sucrose syrup before filtration. Each point represents average values of biological triplicates.

### 2.7. Statistical Analysis

Each measurement was performed in biological triplicate. Statistical tests were performed by GraphPad Prism version 8.0.1 software (San Diego, CA, USA). Differences in *p*-coumaric acid levels between honey samples and sucrose syrup stores from the same colonies were assessed using a paired two-tailed *t*-test, comparing values from individual honey samples across apiaries with the corresponding mean values from sucrose syrup stores from the same colonies. Differences in *p*-coumaric acid levels in sucrose syrup stores sampled in summer, autumn and spring were assessed using a two-way ANOVA and season as fixed factor, followed by Sidak’s multiple comparisons test. Differences in *p*-coumaric acid levels in empty storage combs, brood combs, queen cells and in honey and sucrose syrup samples mixed with propolis were assessed using one-way ANOVA followed by Tukey’s multiple comparisons test. Statistical significance: * *p* < 0.05, ** *p* < 0.01, *** *p* < 0.00.

## 3. Results

In order to evaluate the differences in the *p*-coumaric acid content in various bee products, we quantified its levels in royal jelly, worker diets for larvae of different ages, corbicular pollen, bee bread, honey, empty wax combs and propolis (Figure 1). As previously reported [9], *p*-coumaric acid was undetectable or present only at trace levels in royal jelly (0.05 μg/g on average) whereas it was detected in the worker larval diet, increasing with larval age (0.2 and 0.4 μg/g on average for larvae of 1–3 days and 4–6 days, respectively). Pollen and bee bread contained less than 2 μg/g of *p*-coumaric acid, while honey contained approximately 5 μg/g. Interestingly, empty wax combs contained on average ten times more *p*-coumaric acid than honey (48 μg/g). Propolis exhibited the highest concentrations, exceeding those in honey by over three orders of magnitude (20 mg/g).

To directly compare *p*-coumaric acid levels, we measured its concentration in honey and supplementary sucrose syrup before and after feeding to bees, using paired samples from the same colonies. In the first experiment, frames containing capped sucrose syrup stores were removed from colonies in the spring, following their storage in hives since the previous summer. As expected, freshly prepared sucrose syrup contained no detectable *p*-coumaric acid. However, capped sucrose stores recovered from the colonies contained measurable levels of *p*-coumaric acid, exceeding the concentrations found in honey samples collected from the same colonies prior to syrup feeding (Figure 2A).

To minimize the possibility that the observed increase resulted from mixing of sucrose syrup stores with residual honey by bees during winter, a second experiment was conducted. Capped sucrose syrup combs were removed from colonies shortly after feeding in summer, stored under controlled conditions in the laboratory and *p*-coumaric acid concentrations were monitored over time. The compound was detectable already after two weeks in storage and continued to accumulate throughout autumn and winter in most cases (Figure 2B). We observed that combs from apiaries 3 and 6 were dark, while the comb from apiary 7 was lighter.

Our results indicated that sucrose syrup acquired *p*-coumaric acid during storage in combs. Moreover, syrup stored in darker combs exhibited higher *p*-coumaric acid concentrations than syrup stored in light combs, suggesting a potential association between comb pigmentation and *p*-coumaric acid enrichment (Figure 2C). Since comb pigmentation is closely linked to propolis deposition, we hypothesize that propolis within the combs represents the primary source of *p*-coumaric acid, which diffuses into the bees’ saccharide stores.

To further investigate this hypothesis, we examined the *p*-coumaric acid content of empty wax combs. Most of the wax combs dedicated for saccharide storage are marked by a propolis ring at the cell rims (Figure 3A and [30]), and the thickness of these rings correlates with the apparent darkness of the combs (Figure 3B). We analyzed empty combs built during the same season that had not served for brood rearing, contained no honey or pollen and differed in the thickness of their propolis rings. Combs with thick propolis rings contained significantly higher levels of *p*-coumaric acid than combs with thin or absent rings (Figure 3C). Mechanical removal of the propolis rings resulted in a marked reduction in *p*-coumaric acid content in the empty combs (Figure 3D). The data support the view that propolis incorporated in the comb structure may serve as a source of *p*-coumaric acid for saccharides stored by bees.

Finally, we directly tested whether *p*-coumaric acid from propolis could diffuse into saccharide stores in an in vitro experiment. Elevated levels of *p*-coumaric acid were detected in both honey and sucrose syrup already within one week of incubation with propolis and its concentration remained stable over an extended period of three months (Figure 4A,B). The resulting *p*-coumaric acid concentrations reached 10–20 µg/g that is comparable to the concentrations detected in sucrose stores deposited by bees (compare with Figure 2).

The use of propolis is not limited to combs designated for saccharide storage. Although brood combs lack the prominent propolis rings observed in storage combs, they are characterized by thin internal propolis linings along the cell walls [31,32]. We detected *p*-coumaric acid also in empty brood combs, in amounts comparable to those detected in medium dark empty storage combs (Figure 5A).

In addition to its other roles in promoting honey bee health, dietary *p*-coumaric acid has been shown to suppress ovary development in worker bees [9]. This may be one of the reasons why royal jelly fed to queens contains minimal levels of *p*-coumaric acid [33], as confirmed in our study (Figure 1B). We further examined whether *p*-coumaric acid is present in queen cells after the queen emergence. Indeed, *p*-coumaric acid was detected in empty queen cells, with its concentrations positively correlating with the intensity of queen cell pigmentation (Figure 5B).

## 4. Discussion

This study compared the levels of *p*-coumaric acid in a range of bee products and in supplementary sucrose feeding using the HPLC-HRMS. While freshly prepared sucrose syrup contained only trace amounts of *p*-coumaric acid, syrup stored in combs acquired concentrations similar to those observed in honey. Our results demonstrate that propolis coating the comb surfaces serves as a rich source of *p*-coumaric acid that diffuses into both honey and sucrose syrup during storage. These findings clarify the previously unclear origin of *p*-coumaric acid in honey and provide evidence that supplementary winter feeding does not deprive bees of this key phenolic compound, nor likely other bioactive phytochemicals derived from propolis, that contribute to honey bee health and colony survival.

Honey bees acquire *p*-coumaric acid through the ingestion of pollen and honey [34]. As they lack the ability to synthesize *p*-coumaric acid, they rely entirely on its external sources. While concentrations vary depending on botanical origin, the levels of *p*-coumaric acid in pollen and honey generally fall within a similar range [14,15,16], a finding corroborated by our own data.

As confirmed by our study, the richest source of *p*-coumaric acid and other beneficial polyphenols in the hive is propolis [17,18]. Derived from plant exudates such as tree buds, sap, and resins, propolis exhibits strong antimicrobial and antioxidant properties due to its high content of phenolic acids, flavonoids, terpenes, and other bioactive compounds [35]. Direct consumption of propolis by bees has not been documented in vivo but laboratory studies indicate that its ingestion can enhance immune responses and improve pathogen resistance [36]. Within the hive, bees apply propolis to comb surfaces and hive walls [30], presumably for antiseptic and structural purposes. Our data provide novel and unexpected evidence that bees utilize propolis as an indirect dietary source, whereby *p*-coumaric acid (and likely other polyphenols) diffuses from propolis-coated comb surfaces into the saccharide stores deposited within the comb cells.

Supplementary feeding of honey bees with sucrose syrup or other artificial saccharides has raised concerns about potential *p*-coumaric acid deficiencies that could negatively affect honey bee immunity and overwintering success [5,22]. Our findings contradict this hypothesis. We demonstrate that sucrose-based winter stores, after processing and storage by bees, contain *p*-coumaric acid in concentrations comparable to or even exceeding those in natural honey. These results indicate that the physiological and colony-level impairments observed under sucrose feeding regimes cannot be attributed to a deficiency of *p*-coumaric acid.

The mechanism underlying the enrichment of sucrose stores with *p*-coumaric acid may be explained by two non-mutually exclusive pathways. First, our model proposes that the *p*-coumaric acid originates from propolis that bees deposit on comb surfaces. Second, bees may actively contribute to the enrichment process during nectar or syrup processing by adding salivary gland secretions known to influence the chemical profile of stored honey [37]. Given that bees consume pollen during syrup processing, it is possible that *p*-coumaric acid from the ingested pollen is excreted into the processed syrup. However, as no direct evidence supporting this mechanism has been reported to date, our model appears more plausible.

Although we detected *p*-coumaric acid in empty wax combs, our data suggest that it is not an inherent component of freshly secreted beeswax, as wax foundations did not contain it. Instead, the *p*-coumaric acid signal likely originates from propolis deposits in the combs, from which it diffuses in the saccharide stores over time. Although its water solubility is quite low, we show that *p*-coumaric acid can diffuse into aqueous sucrose solution or honey. This highlights the role of comb associated propolis as a reservoir for phenolic compound enrichment of bee food stores.

Beyond its antimicrobial and antioxidant properties, *p*-coumaric acid is implicated in caste differentiation. As worker bee larvae consume pollen as part of their diet, they are exposed to this acid that has been shown to repress ovary development [9]. In contrast, queen-destined larvae are fed exclusively on royal jelly lacking *p*-coumaric acid [33]. Our study is the first to directly measure *p*-coumaric acid in the diet of worker brood larvae. We confirmed that royal jelly contains only trace amounts, whereas the diet of worker larvae contains *p*-coumaric acid at levels roughly an order of magnitude lower than those in pollen and honey. Despite the importance of avoiding the exposure of *p*-coumaric acid to the developing queens, we detected this acid in queen cells. However, the continuous consumption of royal jelly by the developing larva likely limits its exposure to the *p*-coumaric acid, mitigating any adverse developmental effects. Similarly, while worker brood cells are coated with propolis, the primary source of *p*-coumaric acid for developing worker larvae likely remains dietary pollen [38].

Despite the consistency of our findings across apiaries, some limitations could be considered. The specific chemical composition of propolis varies depending on the botanical sources and local flora that are influenced by geography, climate, and season. As *p*-coumaric acid content in propolis reflects the resin producing plant species available to foragers, the concentrations and diffusion rates observed here may not directly apply to colonies in other regions or ecosystems. In addition, our study focused on colonies maintained under Central European conditions and fed with sucrose syrup in late summer. Other feeding practices and the diffusion of *p*-coumaric acid into alternative types of supplementary feed were not directly assessed in this study. While the *p*-coumaric acid content of propolis and its diffusion rates may vary depending on geographic location and the composition of winter feeds, we expect that the general principles identified here are broadly applicable across different contexts.

In summary, our study provides new insights into the origin and dynamics of *p*-coumaric acid in the hive environment. We demonstrate that both honey and supplementary sucrose stores are enriched with this bioactive compound via comb-associated propolis. This challenges previous assumptions about nutrient deficiencies in artificial feeding regimes and highlights the multifaceted role of propolis as a nutritional source within the hive.

## 5. Conclusions

Our results indicate that propolis is the primary contributor of *p*-coumaric acid enrichment in honey bee saccharide stores. Passive diffusion from propolis-rich combs shapes the phytochemical profile of both honey and supplemental feeds. Consequently, supplementary feed deposited in combs is not devoid of *p*-coumaric acid but acquires levels comparable to those found in natural honey.

## Figures and Tables

**Figure 1 insects-16-01159-f001:**
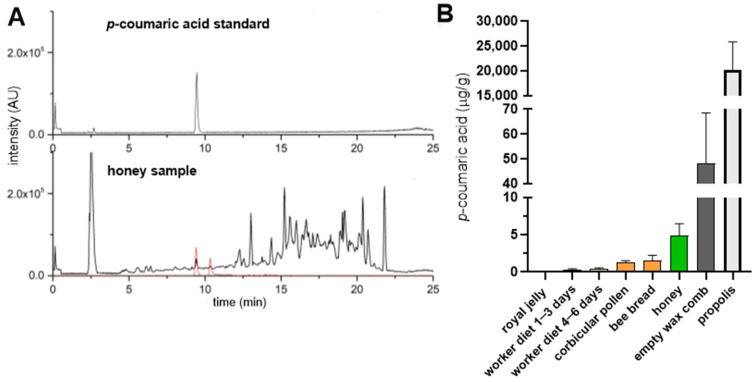
The content of *p*-coumaric acid in various honey bee products. (**A**) HPLC-HRMS chromatogram of *p*-coumaric acid standard solution (top) and honey (bottom, black line). Extracted ion chromatogram indicates ions contributing to *p*-coumaric acid signal (bottom, red line). (**B**) *p*-Coumaric acid levels in various bee products quantified by HPLC-HRMS. Error bars represent standard deviations of biological triplicates.

**Figure 2 insects-16-01159-f002:**
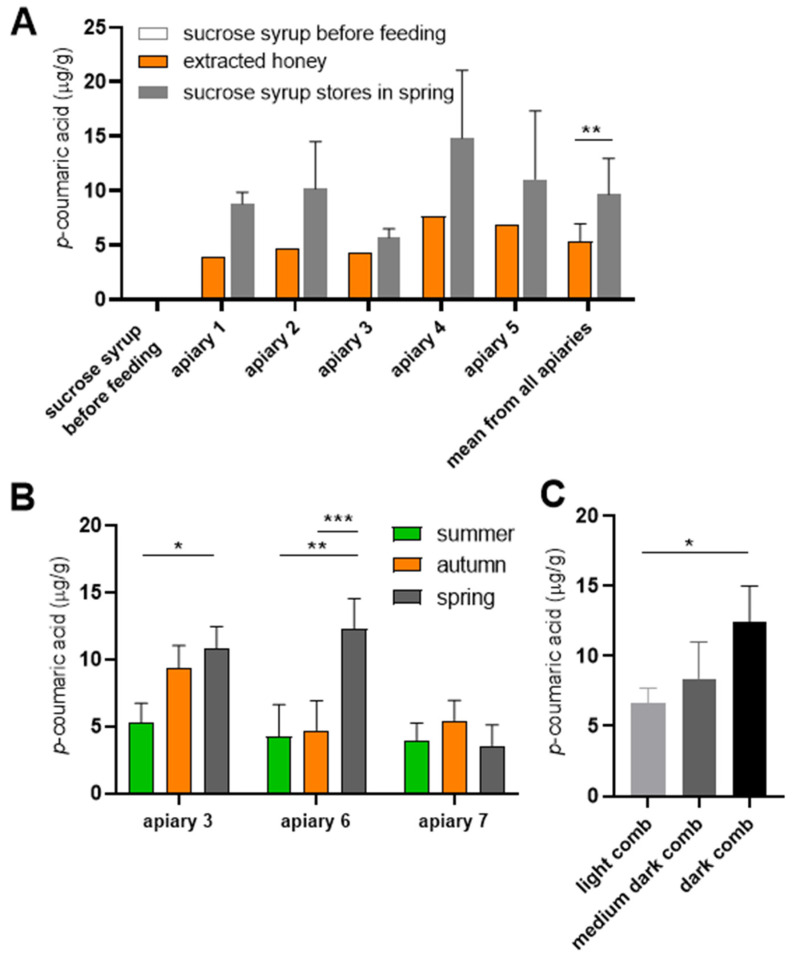
The content of *p*-coumaric acid in honey and sucrose syrup stores originating from the same colonies. (**A**) Honey was extracted in July and stored in glass jars. Sucrose syrup was fed to colonies in August. Empty combs were inserted in suppers just before feeding, freshly capped stores were labelled and left in the colonies until spring when the assays were performed. Apiary 1: Týn and Vltavou; apiary 2: Sedlce; apiary 3: Doubravice; apiary 4: České Budějovice 7; apiary 5: České Budějovice 31. Error bars represent standard deviations of biological triplicates. Significance according to paired two-tailed *t*-test, comparing *p*-coumaric acid values from individual honey samples across all apiaries with the corresponding mean *p*-coumaric acid values from sucrose syrup stores. (**B**) In other colonies the freshly capped sucrose syrup stores were removed in summer, stored at room temperature in the laboratory and the levels of *p*-coumaric acid were analyzed 2 weeks after supplementary feeding in summer, then again in autumn and the spring. Apiary 3: Doubravice; apiary 6: České Budějovice 2; apiary 7: České Budějovice 2B. Significance according to two-way Anova with Sidak’s multiple comparisons test. (**C**) *p*-Coumaric acid levels were analyzed in honey extracted from storage combs of varying colours. Significance according to two-tailed *t*-test. Error bars represent standard deviations of biological triplicates, except in panel A where honey samples from each colonies were analyzed in single measurements. * *p* < 0.05, ** *p* < 0.01, *** *p* < 0.001.

**Figure 3 insects-16-01159-f003:**
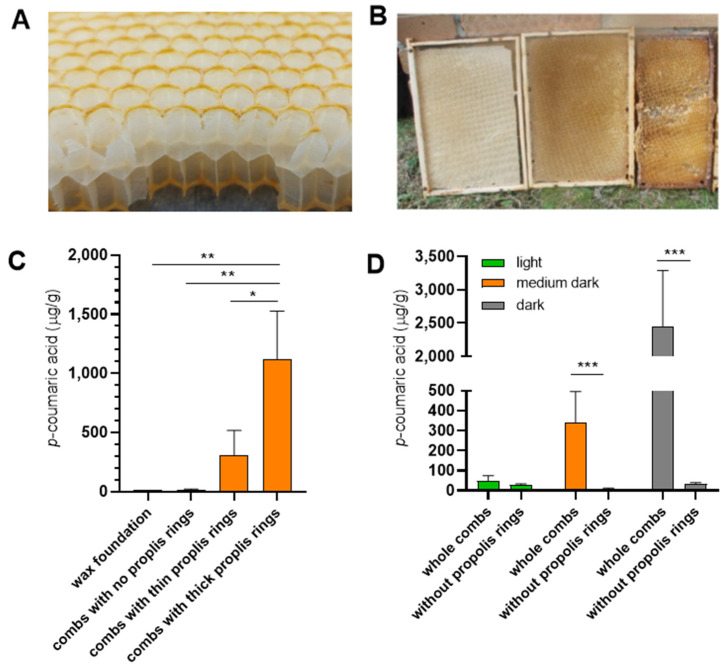
The extraction of *p*-coumaric acid from empty storage combs differing in the presence of propolis rings and in their colour. (**A**) Cross section of a comb dedicated for storage of honey or sucrose syrup. The edges of wax cells are marked by a propolis ring. (**B**) These rings can be of various thickness, making the combs appear darker in colour. Set of storage combs built from wax foundations within the same season. (**C**) *p*-coumaric acid levels in empty storage combs built without the use of wax foundations, differing in the intensities of their propolis rings. Significance according to one-way Anova and Tukey’s multiple comparisons test. (**D**) *p*-coumaric acid levels in combs depicted in (**B**) before and after the wax cell edges with propolis rings were removed. Significance according to one-way Anova and Tukey’s multiple comparisons test. Error bars represent standard deviations of biological triplicates. * *p* < 0.05, ** *p* < 0.01, *** *p* < 0.001.

**Figure 4 insects-16-01159-f004:**
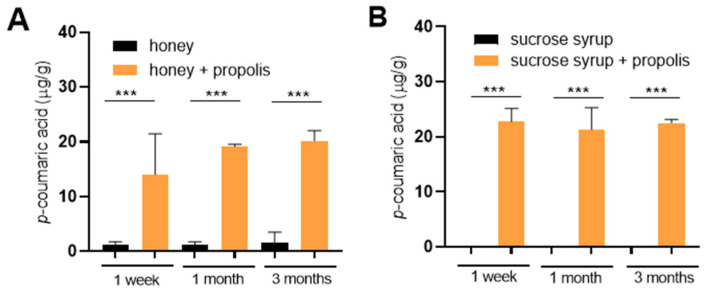
Diffusion of *p*-coumaric acid from propolis to honey and sucrose syrup. Propolis wax mixed with honey (**A**) or sucrose syrup (**B**) and incubated for indicated times in a dark incubator set at 35 °C to mimic hive conditions. Significance according to one-way Anova and Tukey’s multiple comparisons test. Error bars represent standard deviations of biological triplicates. *** *p* < 0.001.

**Figure 5 insects-16-01159-f005:**
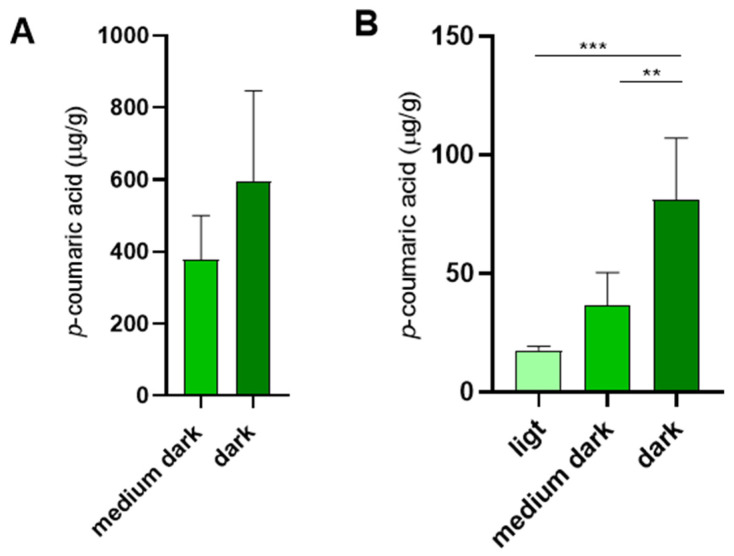
The levels of *p*-coumaric acid in empty brood combs and empty queen cells. (**A**) *p*-coumaric acid levels in empty brood combs of various colours dedicated for rearing young worker bees. (**B**) *p*-coumaric acid levels in queen cells of various colours after queen emergence. Significance according to one-way Anova and Tukey’s multiple comparisons test. Error bars represent standard deviations of biological triplicates (**A**) and quintuplicates (**B**). ** *p* < 0.01, *** *p* < 0.001.

## Data Availability

The original contributions presented in this study are included in the article. Further inquiries can be directed to the corresponding authors.

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
