# Peer review of "Propolis as a Key Source of p-Coumaric Acid Permeating Honey and Sucrose Syrup Stores of Honey Bees"

_insects, 2025, doi:10.3390/insects16111159_

Round 1
Reviewer 1 Report
Comments and Suggestions for Authors
The study is rigorously designed, employs sound experimental methods, and presents data that robustly support its conclusions, marking a clear scientific contribution. It effectively investigates the source and migration pathways of p-coumaric acid within the honey bee colony, revealing that propolis serves as a significant source of this acid—crucial for bee health—in both honey and sucrose syrup stores. This finding challenges the conventional assumption that artificial feeding deprives bees of phenolic acids. The research question is well-defined, the experimental design is appropriate, and the data are reliable. The results offer valuable insights for beekeeping practices and colony health management. The following improvements are recommended:
- Section 2.7, Statistical Analysis: The specific statistical tests used for different metrics should be clearly stated. Furthermore, the significance notation "**** p<0.0001" is unconventional.
- Definition of "empty combs":The manuscript lacks a clear definition of "empty combs" regarding their age and whether they were completely free of residual honey or pollen. It is recommended to supplement the "Materials and Methods" section with a precise definition and description of the processing procedure for these combs.
- It is advisable to have the manuscript polished for language to enhance clarity and readability.
It is advisable to have the manuscript polished for language to enhance clarity and readability.
Author Response
We are grateful to the reviewer for the thorough and thoughtful evaluation of our manuscript and for providing helpful, constructive comments and suggestions. We greatly appreciate the positive assessment of our work. We have made the requested minor revisions, which we believe have further strengthened the paper.
Comment 1: Section 2.7, Statistical Analysis: The specific statistical tests used for different metrics should be clearly stated. Furthermore, the significance notation "**** p<0.0001" is unconventional.
Response 1: We have revised Section 2.7 to clearly specify the statistical tests used for each set of data. In addition, we have replaced the unconventional notation **** with the standard *** format throughout the figures.
Comment 2: Definition of "empty combs":The manuscript lacks a clear definition of "empty combs" regarding their age and whether they were completely free of residual honey or pollen. It is recommended to supplement the "Materials and Methods" section with a precise definition and description of the processing procedure for these combs.
Response 2: We appreciate this suggestion and we have added the important details that were previously missing from the manuscript. We have included a precise definition of “empty combs” in the Methods section (lines 147-153 in the manuscript with highlighted changes). Specifically, we clarify that the empty storage combs used were built in the same season they were analyzed, either with or without the use of wax foundations. They had not been used by the bees for brood rearing nor storage of saccharide stores and pollen. Empty brood combs were combs previously used for brood rearing. Their colour reflected both age and intensity of brood rearing, with darker combs indicating older and/or more intensively used combs. Prior to use, the combs were inspected to ensure the area sampled for p-coumaric acid content was free of honey or pollen. The frames were removed from colonies and stored at 8°C until analysis.
We also included a short remark in the Result section (lines 249-251 in the manuscript with highlighted changes). Moreover, in the legend to Figure 3, we specified whether the analyzed empty combs were built on a wax foundation or constructed without one.
Comment 3: It is advisable to have the manuscript polished for language to enhance clarity and readability.
Response 3: We used in house language service to polish the text. Taking into account all the reviewers’ comments, we believe that the text is now clearer and more easily readable.
Reviewer 2 Report
Comments and Suggestions for Authors
This is an interesting and novel article relating to the feeding of honey bees with sugar syrup and its properties after processing and storage in the hive. It is generally well presented and well written.
Some issues need attention:
Line 84 and the legend for Table ST1: the legend refers to honey bees but the text refers to honey bee products. The legend needs corrected.
The references to the figures start at line 99, referring to Fig 2A. It is unusual to refer to figures in the Results in the Methods section and also not refer to Fig 1 first.
Section 2.7 should specify the various statistical tests used for the different experiments.
The legend given for Fig 1B seems to be in the wrong place and belongs to Fig 2A. Fig 1B then needs a corrected legend.
There is a typo in the labelling of Fig 5B that needs corrected- “ligt” should be “light”.
In Fig 2, not all of the experimental apiaries appear in the sub-figures, but this is not referred to in the text. Some explanation would be helpful.
In Fig 2 as a whole, the position of the marking of the significant differences does not make clear where the differences are- if it is correct, it would at least be clearer to state the significant results in the legend or the main text.
The supplementary materials section needs to be completed- at the moment it is the journal template version that is still there.
In the references “and” should be used before the final author, rather than “a”.
These and various minor points of wording and use of italics or otherwise are marked on the document for the authors’ attention.

Author Response
We are grateful to the reviewer for the thorough and thoughtful evaluation of our manuscript, for providing helpful and constructive comments, and for his/her eagle-eyed attention in spotting overlooked typos and grammatical inaccuracies. We edited all the spelling and grammar mistakes according to the reviewer´s suggestions. We greatly appreciate the positive assessment of our work. We have made the requested revisions, which we believe have further strengthened the paper.
Comment 1: Line 84 and the legend for Table ST1: the legend refers to honey bees but the text refers to honey bee products. The legend needs corrected.
Response 1: We apologize for the confusion. The legend and text have been corrected.
Comment 2: The references to the figures start at line 99, referring to Fig 2A. It is unusual to refer to figures in the Results in the Methods section and also not refer to Fig 1 first.
Response 2: We agree with the reviewer that referring to figures in the Results section is unusual. We thought about this option and we decided to specify Fig. 2A and Fig. 2B in the Methods section to enhance clarity and facilitate understanding of the rather similar, yet complex experimental designs presented in these two panels. By referring to these figures in the Methods, we aim to prevent possible confusion for the reader when interpreting and evaluating the results presented in Figure 2. We therefore propose to retain this approach and keep the references to the specific figures within the methods description.
Comment 3: Section 2.7 should specify the various statistical tests used for the different experiments.
Response 3: We have revised Section 2.7 to clearly specify the statistical tests used for each set of data.
Comment 4: The legend given for Fig 1B seems to be in the wrong place and belongs to Fig 2A. Fig 1B then needs a corrected legend.
Response 4: The reviewer is absolutely correct, and we apologize for any confusion caused. The legends for both figures have now been corrected.
Comment 5: In Fig 2, not all of the experimental apiaries appear in the sub-figures, but this is not referred to in the text. Some explanation would be helpful.
Response 5: Samples for Fig. 2A and 2B were not collected from exactly the same set of apiaries, and therefore the datasets are only partially complementary. We acknowledge that it would have been preferable to collect both types of samples from all apiaries; however, not all apiaries were easily accessible, and we opted to sample from a random subset.
Comment 6: In Fig 2 as a whole, the position of the marking of the significant differences does not make clear where the differences are- if it is correct, it would at least be clearer to state the significant results in the legend or the main text.
Response 6: Unfortunately, we did not notice before submitting the manuscript that all the significance bars in Figure 2 had shifted sideways, making it difficult to interpret the statistical differences. We have now corrected the figure and placed the bars in their proper positions.
Comment 7: The supplementary materials section needs to be completed- at the moment it is the journal template version that is still there.
Response 7: We completed this section.
Comment 8: In the references “and” should be used before the final author, rather than “a”.
Response 8: We have corrected the citations. The errors were caused by using the Czech version of the reference management software.
Reviewer 3 Report
Comments and Suggestions for Authors
I consider the article to be good, well-designed, generally clear and well-written, and an important contribution to the advancement of beekeeping, particularly regarding feeding.
Answer: Does supplemental feeding with sucrose syrup deprive bees of p-coumaric acid?
I believe the findings are novel and have direct practical implications for colony management.
However, I recommend correcting the following:
The introduction is unbalanced; it does not clearly explain the problem, the knowledge gap, or the need for the study. It discusses p-coumaric acid in general and in bees, but does not include information about propolis (this makes it inconsistent with the title) and the other possible sources of coumaric acid that were analyzed, such as: sucrose syrup stores, combs, queen cells, royal jelly, worker larvae.
The statistical analysis should specify which tests were performed; the information contained at the bottom of the figures should be included in the statistical analysis section.
The most serious error I detect is in Figure 2A: the use of a paired-sample t-test with single honey measurements (instead of triplicates) is methodologically incorrect. I recommend presenting the data as observational trends, explaining the limitation in the text and legend.
Improve the structure of the methodology and explain the experimental design to facilitate reader understanding: detail the protocol for inserting empty combs, feeding, marking, and the two sampling schemes (Exp. 1 and Exp. 2).
Reduce the size of the names on the x-axis of the figures.
In the discussion, I suggest beginning the first paragraph by reaffirming the main finding: the supplemental syrup acquires significant levels of p-coumaric acid during storage in combs with propolis.
Include in the discussion the limitations of the study, such as the possible botanical variability of propolis or the generalization of the findings to other regions.
Author Response
We are grateful to the reviewer for the thorough and thoughtful evaluation of our manuscript and for providing helpful, constructive comments and suggestions. We greatly appreciate the positive assessment of our work. We have made the requested revisions, which we believe have strengthened the paper.
Comment 1: The introduction is unbalanced; it does not clearly explain the problem, the knowledge gap, or the need for the study. It discusses p-coumaric acid in general and in bees, but does not include information about propolis (this makes it inconsistent with the title) and the other possible sources of coumaric acid that were analyzed, such as: sucrose syrup stores, combs, queen cells, royal jelly, worker larvae.
Response 1: We thank the reviewer for this insightful comment. We agree that the Introduction could better highlight the knowledge gap and the aims of our study. In response, we have revised the text to include a paragraph on propolis and other matrices analyzed in the study (lines 64-71 in the manuscript with highlighted changes). We have also clearly indicated the knowledge gaps at the end of the second and third paragraphs and, importantly, have specified our study objectives more precisely and broadly in the final paragraph of the Introduction (lines 88-93 in the manuscript with highlighted changes). We hope that these changes now align the Introduction more closely with the title and they clearly justify the rationale for our work.
Comment 2: The statistical analysis should specify which tests were performed; the information contained at the bottom of the figures should be included in the statistical analysis section.
Response 2: We have revised Section 2.7 to clearly specify the statistical tests used for each set of data.
Comment 3: The most serious error I detect is in Figure 2A: the use of a paired-sample t-test with single honey measurements (instead of triplicates) is methodologically incorrect. I recommend presenting the data as observational trends, explaining the limitation in the text and legend.
Response 3: We apologize for the confusion that was created in Figure 2. Unfortunately, we did not notice before submitting the manuscript that all the significance bars in Figure 2 had shifted sideways, making it difficult to correctly locate and interpret the statistical differences. We have now amended the figure and placed the bars in their proper positions.
It should now be apparent in Figure 2A that individual honey measurements were not used directly for statistical tests. Instead, only the values presented in the last two columns of Fig. 2A, which combine data from all apiaries, were compared. A paired t-test was used to compare p-coumaric acid values from individual honey samples across all apiaries with the corresponding mean values from sucrose syrup stores originating from the same hives (i.e., five paired values). This information has now been specified both in the figure 2 legend and in the Methods section.
Comment 4: Improve the structure of the methodology and explain the experimental design to facilitate reader understanding: detail the protocol for inserting empty combs, feeding, marking, and the two sampling schemes (Exp. 1 and Exp. 2).
Response 4: We have carefully reviewed the Methods section and added further details describing the design of Experiment 1 and Experiment 2. We hope that these additions make the methodology clearer and more understandable for readers.
Comment 5: Reduce the size of the names on the x-axis of the figures.
Response 5: In accordance with this suggestion, we have reduced the font sizes in all the graphs. We also shortened the text labels on the x-axis in some graphs.
Comment 6: In the discussion, I suggest beginning the first paragraph by reaffirming the main finding: the supplemental syrup acquires significant levels of p-coumaric acid during storage in combs with propolis.
Response 6: We agree with the reviewer that the Discussion benefits from a summarizing paragraph, which we have now added (lines 321-329 in the manuscript with highlighted changes).
Comment 7: Include in the discussion the limitations of the study, such as the possible botanical variability of propolis or the generalization of the findings to other regions.
Response 7: We thank the reviewer for this valuable suggestion. We have added a paragraph in the Discussion addressing these limitations, including the potential botanical variability of propolis and considerations regarding the generalization of our findings to other regions study (lines 378-388 in the manuscript with highlighted changes).